# OpenReview forum: "Black-Box Detection of LLM-Generated Text Using Generalized Jensen Shannon Divergence"
_ICML.cc/2026/Conference — ICML 2026 regular_

### Official Review · Reviewer_4LuK · 2026-03-09

**Soundness:** 2
**Presentation:** 3
**Significance:** 2
**Originality:** 3
**Overall Recommendation:** 3
**Confidence:** 3

**Summary:**

This paper proposes a black-box detector called SurpMark for detecting AI-generated text. This detector requires comparative analysis using both human and machine reference corpora.  Token surprisals in a text are quantized into states, state transitions are modeled as Markov chains, and GJS scores against human and machine references are computed for comparison. Experimental results show that SurpMark outperforms a few baseline methods on the dataset constructed by the authors.

**Compliance With Llm Reviewing Policy:**

Affirmed.

**Final Justification:**

I still have reservations about the evaluation in this work. Many additional experiments need to be added to the original submission to make the evaluation convincing.

**Key Questions For Authors:**

Please see the weakness section above.

**Limitations:**

yes

**Strengths And Weaknesses:**

Strengths:

1. The idea of summarizing a text into a state-transition matrix looks interesting to me.

2. Experimetal results show that the proposed SurpMark method outperforms quite a few baseline methods.

3. The authors performed comparehensive ablation/comparative analyses in their experiments.


Weaknesses:

1. The proposed SurpMark method relies on both human and machine reference corpora, which may limit its application in real-world scenarios, as we typically lack prior knowledge of the text to be detected and have no access to reference corpora. Furthermore, the authors built reference corpora for each dataset, which makes comparisons with other baselines that do not use specific datasets for training unfair.

2. The proposed method has some key hyperparameters (e.g., the k value for k-means and the threshold τ), which are difficult to set in practice. Regarding the k value, Figure 3 shows that the AUROC score actually fluctuates with the k value.

3. The evaluation was conducted only on the dataset constructed by the authors. Reporting results using existing datasets would be more convincing. Furthermore, since only AUROC is currently used for evaluation, I suggest incorporating other metrics to provide more comprehensive results.

4. As shown in Table 2, the proposed method does not consistently outperform the baseline method on different generators, with its average performance only slightly better than Lastde++ (91.23 vs. 90.04). Statistical significance testing and further analysis are needed.

---

> ### Author Rebuttal · Authors · 2026-03-30
>
> We thank the reviewer for the careful reading and constructive feedback.
>
> **W1**
>
> We thank the reviewer for this comment. SurpMark is a lightweight statistical detector. While it requires a reference corpus, the required scale is modest: Table 4(a) shows that gains become limited once the reference set reaches 100 samples per side, suggesting that **SurpMark does not depend on large-scale reference collection**. This requirement can often be readily satisfied by existing public datasets, such as DetectRL [1]. We suspect that the reviewer's concern about reference-based detectors may partly stem from the paradigm of classification-based detectors, which typically require substantially larger supervised datasets. For example, T5-Sentinel uses OpenLLMText (~340k samples) [2]. Please note in Table 2 and 3, we also **compare against baselines that require additional reference/training data** from both human and LM sides, including R-Detect and FourierGPT, and SurpMark still outperforms them.
>
> Although DNA-GPT, DetectNPR, and DetectGPT do not pre-collect corpora, they require online regeneration for each test sample, which makes them **slow** at test time, and less suitable for deployment. In contrast, SurpMark uses reusable references. As in Line 251, this gives SurpMark a lower reference-per-test budget for our main setting, and it also achieves about 3.4$\times$- 1261$\times$ higher throughput than Lastde++, DNA-GPT, and DetectGPT(Fig 6).
> Also, we would like to clarify that SurpMark does not require a perfectly matched reference set: Table 5 and Table 6 already evaluate cross-domain and cross-generator generalization, and our newly added DetectRL ([table](https://anonymous.4open.science/r/random2-2D43/Detectrl-2.png)) results further confirm strong performance in mixed-domain and mixed-model settings.
>
> Overall, SurpMark shows that **a small amount of offline-collected (or publicly available) reference data** can be effectively translated into **both substantial speedup at test time** and **strong detection performance**.
>
>
>
> **W2**
>
> **Our paper addresses both.** The theory suggests $k^*=\Theta(N^{1/5})$, reflecting a discretization-estimation tradeoff as in Sec 3.2.1: too small $k$ loses distributional differences (human vs. machine), while too large $k$ increases estimation noise. This matches Fig.3, where AUROC improves as $k$ increases to a moderate range and then saturates or declines. Appendix E.2.2 (Table 15) confirms across reference sizes, the empirically best $k$ grows only mildly with $N$, and the ratio $k/N^{1/5}$ stays consistently around $0.8$, matching theory well. Thus, **$k$ is guided by a simple data-size rule**.
>
> For $\tau$, our theory provides a natural default: Appendix E.2.13 explains that $\tau=0$ corresponds to the sign test, since $\Delta \mathrm{GJS}_n$ is exactly the log-likelihood ratio between the machine and human hypotheses. So $\tau=0$ is the **principled Neyman-Pearson decision rule**. Empirically, F1 at $\tau=0$ is close to the oracle F1 (Table 25). Table 26 shows at their respective fixed thresholds, SurpMark attains higher F1 than Lastde++ in all settings.
>
> **W3**
>
> Our data construction follows Fast-DetectGPT and Lastde++: we start from public datasets and construct paired samples for evaluation. The paper already reports results on HC3-Chinese in Fig. 7, where HC3 provides paired HWT and MGT, and SurpMark shows clear gains over baselines.
>
> Our evaluation is not limited to AUROC. In the paper, Table 18 reports TPR at low FPR levels (5%, 1%). Table 25 and Table 26 provide F1 comparisons under fixed thresholds.
>
> To address reviewer's concern, we additionally report results on the DetectRL [1], see [table](https://anonymous.4open.science/r/random2-2D43/Detectrl-2.png). DetectRL proxides a more challenging test of cross-domain and cross-generator robustness: each domain-specific set mixes source models and each model-specific set mixes domains. We report AUROC and TPR@FPR=5%. The results show that our method consistently outperforms existing baselines.
>
>
>
>
>
> **W4**
>
> Lastde++ is a strong baseline on open-source generators (Table 2) but less competitive in more challenging settings.
>
> We note Table 2 reports AUROC averaged across three datasets, while per-dataset results (mean over three runs with std) are in Tables 12-14. Since the open-source generators in Table 2 are relatively small, the task is easier and the gap between strong methods is limited. In more challenging settings, closed-source models (78.32 vs. 61.15 in Table 3), non-English text (Fig. 7), SurpMark shows much clearer gains over Lastde++. Our added DetectRL and MixSet results (see response to W3 for reviewer dtXv) show that SurpMark is clearly better than Lastde++ in mixed-domain/mixed-model and human-paraphrased / LM-polished settings.
>
>
>
> [1] DetectRL: Benchmarking LLM-Generated Text Detection in Real-World Scenarios, NeurIPS24
>
> [2] Token Prediction as Implicit Classification to Identify LLM-Generated Text, EMNLP23

---

> > ### Author Rebuttal · Reviewer_4LuK · 2026-04-03
> >
> > Some of my concerns have been addressed. However, I still have some reservations regarding the evaluation: 1) In addition to the self-built dataset, several existing datasets built by other people should be tested, such as Kaggle competition datasets, Fast-detectgpt datasets, etc. 2) For this task, TPR at low FPR is a more meaningful metric, and I suggest the authors also report the results for this metric. 3) Using different reference corpora can actually have a significant or slight impact on detection performance, which increases the difficulty of method deployment.

---

> > > ### Author Response · Authors · 2026-04-04
> > >
> > > ## Existing Datasets
> > >
> > >
> > > As requested by the reviewer, we add results on the Kaggle DAIGT V2 Train Dataset, which represents a challenging mixed-generator setting with texts generated by more than 15 models. Across multiple runs, SurpMark consistently outperforms baselines.
> > >
> > > | Method         | AUROC   (std)     | TPR@FPR=5% (std)  |
> > > |----------------|--------------|--------------|
> > > | Lastde++       | 88.37 (2.45) | 78.00 (4.06) |
> > > | Fast-DetectGPT | 88.75 (2.49) | 81.33 (4.16) |
> > > | SurpMark       | **96.17** (1.56) | **85.78** (1.39) |
> > >
> > >
> > > We would like to clarify Tables 12 and 13 of the paper include results evaluated on subsets of the Fast-DetectGPT dataset. For GPT-2-large, GPT-j-6b, GPT-Neo-2.7B, GPT-Neo-20B, and OPT-2.7B, the XSum and WritingPrompts subsets of the Fast-DetectGPT dataset contain sufficient samples, and the corresponding comparison results have been presented in Tables 12,13. We now report TPR@FPR=5% for these settings. For SurpMark, k is chosen as the value that achieves the best AUROC for the corresponding setting in Tables 12 and 13. These results are consistent with trend in Table 2: on earlier and smaller generator models, the performance gaps among methods are narrower, so the improvement margin is limited.
> > >
> > > | TPR@FPR=5%     | SurpMark | Fast-DetectGPT | Lastde++ |
> > > |---------------------|----------|----------------|----------|
> > > | GPT-j-6b (XSum)     | 55.33    | 44.00          | 52.00    |
> > > | GPT-j-6b (WP)       | 89.33    | 82.00          | 88.67    |
> > > | GPT-Neo-2.7B (XSum) | 66.67    | 48.67          | 64.00    |
> > > | GPT-Neo-2.7B (WP)   | 90.00    | 84.67          | 92.00    |
> > > | GPT-Neo-20B (XSum)  | 40.33    | 36.67          | 39.33    |
> > > | GPT-Neo-20B (WP)    | 81.33    | 72.00          | 80.67    |
> > > | GPT-2-large (XSum)  | 90.00    | 84.67          | 92.00    |
> > > | GPT-2-large (WP)    | 99.33    | 99.33          | 100.00   |
> > > | OPT-2.7B (XSum)     | 68.67    | 50.00          | 60.67    |
> > > | OPT-2.7B (WP)       | 86.67    | 76.67          | 90.67    |
> > > | Mean  | **76.77**    | 67.87          | 76.00    |
> > >
> > >
> > > In addition, we extended the comparison to the larger-model subsets of the Fast-DetectGPT dataset(GPT-3.5-Turbo and GPT-4). Because the number of samples for larger-model subsets is limited, we evaluated them under a mixed setting. For each model, we evenly mixed samples from PubMed, WritingPrompts, and XSum to form a single dataset (now 450 samples per side). We sampled 300 samples from each side as reference and used the remaining 150 for testing.
> > >
> > >
> > > GPT-3.5-Turbo
> > >
> > > | Method         | AUROC (std) | TPR@5%FPR (std) |
> > > |----------------|-------|-----------|
> > > | SurpMark       | 95.01 (0.45)| 77.17  (2.69)   |
> > > | Lastde++       | 90.23 (1.21) | 70.00   (2)  |
> > > | Fast-DetectGPT | 93.62 (0.74) | 77.78 (1.02)     |
> > >
> > > GPT-4
> > >
> > > | Method         | AUROC (std) | TPR@5%FPR (std) |
> > > |----------------|-------|-----------|
> > > | SurpMark       | 88.89 (0.58)| 60.00  (2) |
> > > | Lastde++       | 81.03 (0.2)| 44.56  (0.51)   |
> > > | Fast-DetectGPT | 86.17 (1.02)| 50.00  (3.56)   |
> > >
> > >
> > >
> > >
> > > ## Deployment
> > >
> > > We agree that using different reference corpora can affect detection performance. However, we respectfully view this not as a prohibitive deployment barrier, but as a **manageable deployment trade-off**. In practice, the key issue is not whether the detector score distribution changes, but whether such changes lead to a **substantial threshold re-calibration burden**.
> > >
> > > First, our results suggest **SurpMark is reasonably robust to the choice of reference corpus**. In Table 5, using self-ref as the in-domain reference, **19/24 cross-domain settings remain within a ±2% relative change**, indicating SurpMark captures transition patterns that transfer well across different corpora.
> > >
> > > Second, we would like to emphasize that an often overlooked deployment cost of baselines lies in threshold selection. For example, Fast-DetectGPT uses empirically chosen operating thresholds , and in our tests we observe its score distribution changes with input length ([figure](https://anonymous.4open.science/r/random2-2D43/fastdetectgpt-score-distribution.png)). This indicates that **careful threshold calibration may be needed when transferring across settings, introducing a potentially expensive deployment overhead because it requires both additional calibration samples and repeated computation to validate a reliable operating threshold.**
> > >
> > > Unlike heuristic threshold selection in prior methods, SurpMark has a theory-motivated default threshold ($\tau = 0$), which reduces reliance on manual tuning.
> > >
> > > Overall, even when reference corpus changes, SurpMark maintains a favorable deployment trade-off by replacing repeated heuristic threshold tuning with a lightweight offline reference-update and a **much more stable decision rule**, making deployment more predictable.
> > >
> > > UPD: We note our added OOD results (to Reviewer dtXv) are relevant to your concern about reference-corpus shift: in the added domain-shift settings, SurpMark remains much stronger than the baselines.

---

### Official Review · Reviewer_dtXv · 2026-03-12

**Soundness:** 3
**Presentation:** 3
**Significance:** 2
**Originality:** 3
**Overall Recommendation:** 3
**Confidence:** 4

**Summary:**

The paper proposes SurkMark, a way of detecting LLM generated text. SurpMark works by discretizing surprisals then modeling a markov chain progressing through the text. Detection is done by thresholding the difference between the Jensen Shannon Divergence between the transition matrix computed from the input text against one modeled for human text and one modeled for LLM text. SurkMark outperforms baselines and other black-box detectors across various open source models and corpuses.

**Compliance With Llm Reviewing Policy:**

Affirmed.

**Key Questions For Authors:**

* How does the problem change in non black box setting? What happens if the proxy LM equals the generating LM in this case? Would love to see that ablation
* How does this change at scale? Does a larger proxy LM work better than a smaller proxy? Smaller models typically have much more predictable behavior and lower diversity in their responses. Would love to see experiments with DeepSeek or Kimi.

**Limitations:**

The authors mention that there are no societal consequences of the work to be discussed, but I heavily disagree. The authors report AUROC as a number, which are typically higher numbers than accuracy or F1 and a misinformed viewer might read a high numbers and assume a high trustworthiness of the classifier. This could result in unwarranted uses of this project - for instance, teachers flagging students for plagiarism on essays. I would prefer if the authors write a little bit about how a detection from the classifier does not imply that the text is 100% AI generated.

**Strengths And Weaknesses:**

Strengths:
* The idea is novel. I like the intuition behind modeling the transition matrix between tokens, and the insight that LLMs typically have a low surprisal token after a high surprisal token is clean and intuitive.
* Experiment rigor is good. Ablations are done across proxy LM and generating LM, corpuses and several baselines are presented in the comparison tables for each of these ablations.
* The paper is well presented with nice diagrams and comprehensive analysis. The equations are clearly written and explanation is done well.

Weaknesses:
* I would've liked to see some analysis reporting F1 score in addition to AUROC. I believe a strong application of a method like this is filtering pretraining data, for which you'd have to tune $\tau$ accordingly. Ablating it will be quite expensive for pretraining, so it would be useful to see at a glance how the model performs for a fixed $\tau$
* The domain generalization doesn't look as good as the authors claim. Seeing a lot of 70-80s across Table 5. I believe this is the most important metric to capture for LLM detection methods, because it's near impossible to have a fully representative corpora of text to train on. It's also unclear whether this method is applicable to larger models as the authors do not report this metric with text generated or proxied by frontier models.
* While the authors look into paraphrasing, I think there is much analysis left to do. Does the answer change if a human is the one paraphrasing? How does it change if part of the text is written by a human and part is written by an LLM?
* Human baseline is missing - how difficult is the corpora that the authors collected? Can a human detect whether the text is LLM generated?

---

> ### Author Rebuttal · Authors · 2026-03-30
>
> We thank the reviewer for the constructive feedback. Below we address these points in detail.
>
> **W1**
>
> As noted in Line 416 and Appendix E.2.13, we already study fixed-threshold performance and the choice of $\tau$. $\tau=0$ is not an ad hoc tuning knob, but the natural decision rule induced by our GJS-gap statistic. Empirically, Table 25 shows this choice achieves F1 close to the oracle threshold, and Table 26 shows under fixed thresholds, SurpMark consistently outperforms Lastde++. Among baselines examined, only Lastde++ suggests a specific threshold value in the paper, based on empirical observation. This makes our principled $\tau=0$ appealing for deployment settings where threshold sweeps are costly. We have added an additional F1 analysis ([table](https://anonymous.4open.science/r/random2-2D43/F1.png)). SurpMark consistently outperforms Lastde++ in all settings.
>
>
>
> **W2**
>
> Table 5 measures robustness to reference-domain shift, rather than absolute AUROC alone. Using self-ref as the in-domain baseline, we find 19/24 cross-domain settings remain within ±2% relative change. To further address generalization, we add results on DetectRL [1] ([table](https://anonymous.4open.science/r/random2-2D43/Detectrl-2.png)), where SurpMark consistently outperforms prior baselines in mixed-domain/mixed-model settings.
>
> For frontier models, Table 3 already reports results on text generated by 3 frontier closed-source models, where SurpMark shows a clear advantage over baselines. This is consistent with the added DetectRL results, where SurpMark substantially outperforms baselines on Claude-instant **(80.92 vs. 37.16/39.11)**. We cannot add frontier-proxy experiments due to API limitations; we explain this constraint in detail in our response to Q2.
>
> [1] DetectRL: Benchmarking LLM-Generated Text Detection in Real-World Scenarios, NeurIPS24
>
> **W3**
>
> We added new experiments using MixSet [2], a benchmark for human-revised and mixed-authorship text.
>
> For human paraphrasing, we compare HWT vs. human-revised MGT ([table](https://anonymous.4open.science/r/random2-2D43/human-vs-human-revised.png)). SurpMark consistently outperforms baselines at both sentence and token levels.
>
> For mixed-authorship, we report HWT vs. Mixed, Mixed vs. MGT, and HWT vs. MGT ([table](https://anonymous.4open.science/r/random2-2D43/MixSet-Mixed.png)), where the Mixed is originally human-written and then polished by GPT-4. In all settings, the reference are pure HWT and MGT.
> The results suggest that distinguishing human-written text, that has been partially LM-polished, from pure HWT is the harder problem for existing methods, and this is where our method shows the largest advantage.
>
> [2] LLM-as-a-Coauthor: Can Mixed Human-Written and Machine-Generated Text Be Detected? NAACL24, Findings
>
>
> **W4**
>
>
> Conducting a controlled human-subject study during the rebuttal period is not feasible because it requires both additional time and a carefully designed annotation/ethics protocol.
>
>
> We have added results on DetectRL, a public benchmark for text detection. Our method remains the strongest on it, suggesting that the results are not simply due to an easy or overly curated corpus. Separately, to provide some qualitative intuition about the collected corpus difficulty, we include a few examples [table](https://anonymous.4open.science/r/random2-2D43/example.png).
>
> To the best of our knowledge, among the 9 papers corresponding to our detector baselines, only 1 ([3]) includes a human baseline, while the other 8 do not.
>
> [3] GLTR: Statistical Detection and Visualization of Generated Text
>
> **Q1**
>
>
> We have added this ablation ([table](https://anonymous.4open.science/r/random2-2D43/white-box.png)) and observe that performance is very high in the white-box setting. This observation is in line with prior work Lastde++, which reports detection becomes easier in the white-box setting. Importantly, SurpMark still remains the best in more practical black-box setting.
>
> **Q2**
>
> Testing with frontier proxies is currently not feasible, because current flagship closed-source APIs provide continuations and corresponding log-probs, but they do not expose the prompt-token log-probs required by our proxy model. Running frontier models locally is beyond our compute budget.
>
> We note that Fig. 5(c) already studies proxy choice. Following Fast-DetectGPT and Lastde++, we focus on lightweight proxy LMs standard in prior work. We extend this analysis to **larger proxies** ([table](https://anonymous.4open.science/r/random2-2D43/proxy.png)). While larger proxies improve baselines, SurpMark consistently performs better across all proxy models and remains strong even with small proxies.
>
> **Limitations**
>
> We are happy to revise our statements on societal consequences. Detection output should be treated only as a probabilistic signal, not definitive evidence, and should not be used as the sole basis for high-stakes decisions such as plagiarism accusations. We will clarify this in the paper.

---

> > ### Author Rebuttal · Reviewer_dtXv · 2026-04-04
> >
> > Thank you for your response. I'm unfortunately still not convinced by the performance out-of-domain generalization - it does not look that much stronger than the baseline (asides from HWT vs. Mixed). I believe that this is the most important metric to look for, so I don't think the paper is ready for publication yet.

---

> > > ### Author Response · Authors · 2026-04-05
> > >
> > > We thank the reviewer for raising this point. We realize that our initial response emphasized mixed-domain evaluation, whereas the reviewer's main concern is more specifically about out-of-domain reference generalization. To directly address reviewer's concern, we added a new suite of out-of-distribution (OOD) experiments on DetectRL covering four experiment families: domain shift, cross-source corruption, character/word perturbation, and paraphrasing. **Across all 17 OOD settings, SurpMark consistently outperforms the strongest baseline, with average gains of 9.45 AUROC and 17.04 TPR@FPR=5%**.
> > >
> > > #	Domain-Shift OOD
> > >
> > > [[table](https://anonymous.4open.science/r/random2-2D43/domain-shift-ood.png)]
> > >
> > > We first evaluate a domain-shift setting, where the reference and test sets come from different domains under a mixed-generator setup (Llama-2-70B, Claude-Instant, and GPT-3.5-Turbo). This is more challenging than Table 5 because the generated texts come from larger frontier models and multiple generators, yielding a more diverse reference distribution and a harder OOD problem. Here, self-ref uses references from the same target domain, while WP-as-ref, arxiv-as-ref, and yelp-as-ref use references from a different domain, making them genuine cross-domain OOD cases. Importantly, SurpMark remains robust under these cross-reference shifts and does not collapse when the reference domain changes.
> > >
> > > The gains in Table 5 are smaller because the generated texts come from smaller models hich makes the detection problem easier overall and therefore reduces the margin for improvement over the baselines. By contrast, the new DetectRL results address the reviewer's concern more directly. **In genuine out-of-domain reference settings, SurpMark improves over the strongest baseline by an average of 13.26 AUROC and 11.06 TPR@FPR=5%**.
> > >
> > >
> > >
> > >
> > > # Cross-Source Corruption OOD
> > >
> > > [[table](https://anonymous.4open.science/r/random2-2D43/cross-source-corruption-ood.png)]
> > >
> > > We consider a practically important setting in which documents are not purely human-written or purely LLM-generated, but instead contain sentence-level mixtures from both sources. This is a practical setting, as real-world documents are often hybrids of human writing and LLM-assisted revisions rather than clean single-source texts. To simulate this scenario, we construct a cross-source corruption setting in which the reference sets are pure human-written and LLM-generated texts, while the test sets are partially corrupted at the sentence level with the opposite source. Specifically, human test samples are injected with a subset of LLM-generated sentences, and LLM test samples are injected with a subset of human-written sentences.
> > >
> > >
> > >
> > >
> > > # Character-Word Perturbation OOD
> > >
> > > [[table](https://anonymous.4open.science/r/random2-2D43/character-word-perturbation-ood.png)]
> > >
> > > We consider character- and word-level perturbations as a controlled form of surface-level distribution shift, including spelling noise, character edits, and word substitutions. In this setting, the reference sets remain clean and source-pure, while the test sets are constructed by applying character-level, word-level, or mixed perturbations to both human-written and LLM-generated texts. This creates an OOD shift by modifying the surface form of the test texts through perturbations.
> > >
> > > # Paraphrase OOD
> > >
> > > [[table](https://anonymous.4open.science/r/random2-2D43/paraphrase-ood.png)]
> > >
> > > We consider a paraphrase-based OOD setting in which the reference sets contain pure human-written and LLM-generated texts, while the test sets contain paraphrased versions of those texts. This creates an OOD shift by altering surface form while largely preserving meaning and source identity. We consider four paraphrasing settings: polish uses LM-based fluency rewriting, Back-translation uses round-trip translation, DIPPER uses a dedicated paraphrasing model to generate semantically equivalent rewrites with stronger lexical and syntactic variation, and Mixed combines all three transformations.
> > >
> > > Overall, **SurpMark consistently outperforms the strongest baseline across all 17 settings, with an average gain of 9.45 AUROC and 17.04 TPR@FPR=5%**. The gains remain positive in every experiment family (summarized below), and are particularly strong under cross-source corruption and paraphrase OOD, two of the most practically relevant and challenging deployment scenarios. These results directly address the reviewer’s concern by demonstrating that SurpMark generalizes robustly across diverse OOD conditions, rather than only under a narrow or favorable setup.
> > >
> > > | Experiment family | # Settings | Avg. AUROC gain | Avg. TPR@FPR=5% gain |
> > > |---|---:|---:|---:|
> > > | Domain-shift OOD  | 6 | 13.26 | 11.06 |
> > > | Cross-source corruption | 4 | 10.52 | 17.50 |
> > > | Character-word perturbation | 3 | 1.52 | 8.67 |
> > > | Paraphrase-induced OOD shift | 4 | 8.63 | 31.83 |
> > > | **Overall OOD** | **17** | **9.45** | **17.04** |

---

### Official Review · Reviewer_pYxU · 2026-03-13

**Soundness:** 4
**Presentation:** 3
**Significance:** 3
**Originality:** 3
**Overall Recommendation:** 5
**Confidence:** 4

**Summary:**

This paper proposes a new method for detecting LLM-generated text with only black-box access to LLM outputs. The method aims to take advantage of the observation that a token with high surprisal is immediately followed by a low-surprisal token more frequently in LLM-generated text than in human-generated text. The method works by first modeling surprisal transitions (gotten with a proxy LLM) in a reference human text corpus and a reference LLM text corpus with Markov chains. It then performs a likelihood-free hypothesis test with a test statistic derived from the Generalized Jensen-Shannon Divergence between a test example's surprisal transitions and the surprisal transitions of the two reference corpora. The proposed method often outperforms existing methods when evaluated against 13 existing detection methods on standard benchmarks for detecting generated text from 12 models.

**Compliance With Llm Reviewing Policy:**

Affirmed.

**Final Justification:**

The authors satisfactorily answered my questions with additional experiments during the rebuttal. I am keeping my positive score.

**Key Questions For Authors:**

1. How do you think your your method will perform in a more challenging evaluation setting without such closely paired examples of human/reference text?
2. A few pages have vertical space removed, impacting readability; I recommend allowing the standard spacing.

**Limitations:**

yes

**Strengths And Weaknesses:**

Strengths:
1. The paper identifies a novel difference between human text and LLM-generated text: in the latter, tokens with high surprisal are more often followed by tokens with low surprisal.
2. The method strongly outperforms existing methods at detecting text from large closed-source models, with analysis showing that this is due to the fact that large models have nearly no difference from human text in marginal surprisal distribution but enough difference in the distribution of surprisal differences.
3. The method is more efficient than many of the best-performing alternatives because it builds the reference statistics once instead of generating a neighborhood per input at test time.
4. The paper provides theoretical analysis to guide the method's design decisions like the number of bins when discretizing surprisal and the choice of test statistic.


Weaknesses:

I see no major weaknesses of the paper. The results on smaller open models in Table 2 are not consistently much superior than baselines (often differing by less than 1 point of mean AUC), but this is made up for by the fact that the method performs significantly better than baselines at detecting text from closed models in Table 3.

---

> ### Author Rebuttal · Authors · 2026-03-30
>
> We thank the reviewer for the positive and detailed assessment of our paper. We greatly appreciate the reviewer's recognition of the novelty of our core insight.
>
> **Q1**
>
> To address this concern, we additionally evaluated our method on the public DetectRL [1] benchmark [table](https://anonymous.4open.science/r/random2-2D43/Detectrl-2.png), which is a more challenging setting where the samples are not paired human/reference texts. Despite this, our method still consistently outperforms prior baselines across different source models and domains. This suggests that our method does not rely on closely paired examples and remains effective in more realistic evaluation settings where such pairing is unavailable.
>
> [1] DetectRL: Benchmarking LLM-Generated Text Detection in Real-World Scenarios, NeurIPS24
>
> **Q2**
>
> Thank you for the suggestion. We will restore the standard vertical spacing to improve readability in the final version.

---

> > ### Author Rebuttal · Reviewer_pYxU · 2026-04-04
> >
> > All of my concerns are fully resolved and I keep my positive score. Thanks for running the additional experiments on DetectRL.

---

> > > ### Author Response · Authors · 2026-04-07
> > >
> > > Thank you very much for your kind acknowledgment. We sincerely appreciate your time and consideration.

---

### Official Review · Reviewer_KN4D · 2026-03-16

**Soundness:** 4
**Presentation:** 3
**Significance:** 3
**Originality:** 3
**Overall Recommendation:** 5
**Confidence:** 3

**Summary:**

The paper focuses on the issue of detection of machine-generated text. The authors observe that a difference between machine-generated and human-written text is the distribution of transitions between token-to-token surprisal rates (e.g., what comes before and after a particularly high-surprisal token?) Using this insight, they design a method, SurpMark, that constructs transition matrices between buckets of surprisal and evaluates the Jensen-Shannon gap between the observed transition matrix and prehoc human text / machine text transition matrices. They provide empirical support for the effectiveness of SurpMark and prove several results that aid in the configuring of the hyperparameters for the method.

**Compliance With Llm Reviewing Policy:**

Affirmed.

**Final Justification:**

The rebuttal addressed my primary concern, which was about whether results were influenced by pretraining data contamination. Given this, I've raised my soundness 3->4 and overall score 4->5. I think this paper makes a meaningful contribution to the problem of LM-generated text detection.

**Key Questions For Authors:**

Q1. Can you provide an analysis on a text-model pair where the human-written text is more recent than the model training?

Q2. Can you elaborate further on the point in line 230-233: what do you mean here by balancing? I am not entirely sure how you derived k* from these terms.

**Limitations:**

yes, although I think the limitations section could mention how the method performs better at longer text lengths.

**Strengths And Weaknesses:**

S1. The paper proposes a method for an important problem, and provides thorough empirical and theoretical support for the method. The analysis is thorough and provides additional recommendations for how best to use the method.

S2. The key ideas are quite clever and clearly effective. I think the discretization of transition state to enable a fixed-size transition matrix across variable sequence length is particularly clever and well-motivated.

W1. My largest concern is about the impact of pretraining contamination for the human sequences. There exists some interesting work ([Shi et al 2024](https://openreview.net/forum?id=zWqr3MQuNs)) that shows that text seen in pretraining exhibits smoother per-token likelihood under that model. All of the datasets chosen here are relatively old, and some (e.g. XSum) were based on publicly available data. Is it possible that some of the difference you're detecting is just a feature of the human references appearing in the pretraining data?

---

> ### Author Rebuttal · Authors · 2026-03-30
>
> We sincerely thank the reviewer for the positive evaluation and encouraging feedback. Below, we respond to the weaknesses and questions in detail.
>
> **W1 & Q1**
>
> As requested by the reviewer we have performed an experiment where the HWT dataset is newer than the LLM. We added results on Dolly, whose human-written responses were released in 2023. In particular, when using GPT2-Large (2019) as the proxy model, Dolly thus provides a useful recent-human / older-proxy setting. We have shown the reults in [table](https://anonymous.4open.science/r/random2-2D43/dolly.png) where we present the AUROC results.  We note that AUROC using GPT2-Large is comparable to Qwen-2-0.5B-Instruct (2024) which is a newer model. Interestingly the AUROC for GPT2-Large a bit higher which suggests that our results are not an artifact of a feature of human reference appearing in the pre-training data.
>
> To further examine this, we measured the smoothness of token surprisal sequences using the mean absolute first-order difference, where lower values indicate smoother token dynamics. In [table](https://anonymous.4open.science/r/random2-2D43/smoothness.png), MGT remains much smoother than HWT in token dynamics, consistent with the observation in [Lastde](https://openreview.net/forum?id=vo4AHjowKi) (Fig 1(a)). More importantly, HWT on Dolly is slightly smoother proxied by Qwen-2-0.5B-Instruct than proxied by GPT2-Large. This trend is consistent with prior findings that pretraining exposure makes human-written text smoother in token dynamics. In our setting, this smoother trajectory makes human text more LM-like, which would make detection harder, not easier. Nevertheless, regardless of which proxy model is used, the smoothness of HWT remain markedly different from those of MGT, and this persistent discrepancy is what enables our detector to work effectively.
>
> Overall, these two results suggest that pretraining contamination alone is unlikely to explain our findings; if anything, it would make our reported performance a conservative estimate.
>
> **Q2**
>
> We apologize for the lack of clarity. Here, ``balancing'' means selecting $k$ to trade off the discretization error $O(1/k)$ and the dominant estimation error $O(k^{3/2}/\sqrt{N})$ (up to logarithmic factors). Since the former decreases with larger $k$ while the latter increases, we choose $k$ by minimizing their sum, which gives $k^\*=\Theta(N^{1/5})$. In other words, $k$ should be large enough to distributional differences (human vs. machine), but not so large that the estimated transition matrix becomes too noisy.

---

> > ### Author Rebuttal · Reviewer_KN4D · 2026-04-04
> >
> > Thank you for the additional experiments! This addresses my concern, and I've raised my score accordingly.
> >
> > >  In our setting, this smoother trajectory makes human text more LM-like, which would make detection harder, not easier.
> >
> > I think it would be useful to mention this in the limitations--- as, if I'm understanding this correctly, this also implies that the false positive rate may be slightly higher for texts appearing in the pretraining corpora.

---

> > > ### Author Response · Authors · 2026-04-07
> > >
> > > Thank you for the thoughtful follow-up and for raising your score. This is a helpful point, we will make it explicit in the limitations section of the revised paper.

---

### Decision · Program_Chairs · 2026-04-30

**Decision:**

Accept (regular)

**Comment:**

The paper studies the important problem of black-box detection of machine-generated text. They propose a novel approach centered around  the difference in distributions of the tokens preceding/following a 'surprising' token between machine and human generated text. As Reviewer KN4D notes, some of the ideas are quite clever.

There is both thorough theoretical analysis as well as empirical validation compared to a large number of baselines. Furthermore, the authors have thoroughly responded to reviewer concerns including evaluation on additional datasets (e.g. OOD settings) and clarified that they report additional metrics in addition to AUROC (e.g. Table 18, 25, 26)

I think this is an exciting paper and I strongly support acceptance.